# Use of Endoscopic Scraper and Cell Block Technique as a Replacement for Conventional Brush for Diagnosing Malignant Biliary Strictures

**DOI:** 10.3390/cancers14174147

**Published:** 2022-08-27

**Authors:** Akihisa Kato, Hiroyuki Kato, Itaru Naitoh, Kazuki Hayashi, Michihiro Yoshida, Yasuki Hori, Kenta Kachi, Go Asano, Hidenori Sahashi, Tadashi Toyohara, Kayoko Kuno, Yusuke Kito, Satoru Takahashi, Hiromi Kataoka

**Affiliations:** 1Department of Gastroenterology and Metabolism, Graduate School of Medical Sciences, Nagoya City University, 1 Kawasumi, Mizuho-cho, Mizuho-ku, Nagoya 467-8601, Japan; 2Department of Experimental Pathology and Tumor Biology, Graduate School of Medical Sciences, Nagoya City University, Nagoya 467-8601, Japan

**Keywords:** endoscopic scraper, cell block, biliary strictures, conventional brush

## Abstract

**Simple Summary:**

Brush cytology remains the primary method used worldwide for diagnosing malignant biliary strictures, despite its low sensitivity. Although the endoscopic scraper, the simplicity of which is comparable to that of a conventional brush, has been used mainly in Japan, it has yet to gain popularity on a global scale. The purpose of this study was to evaluate the diagnostic performance of the endoscopic scraper by comparing diagnostic yields and the number of collected cells using quantitative digital image analysis. Our study revealed that the endoscopic scraper and cell block method achieved higher sensitivity than the brush with the cell block method, and showed that the number of cells on the cell block sections obtained by the endoscopic scraper were significantly higher than those obtained using the brush. Given its ease of use and high sample acquisition capability, the endoscopic scraper could replace brush cytology for diagnosing malignant biliary strictures.

**Abstract:**

Histological evidence is essential for diagnosing malignant biliary strictures. However, conventional brush cytology remains the primary method used worldwide, despite its low diagnostic sensitivity and accuracy, as it is technically easy, rapid, and cost-effective. The aim of this study was to evaluate the diagnostic performance of a recently introduced endoscopic scraper, the simplicity of which is comparable to that of a conventional brush, by comparing diagnostic yields and the number of collected cells. The sensitivity of the endoscopic scraper when using the cell block technique was significantly higher than when using brush cytology or a brush with the cell block technique (53.6% vs. 30.9%, *p* < 0.001; 53.6% vs. 31.6%, *p* = 0.024, respectively). Quantitative digital image analysis of cell block sections revealed that the median number of cells obtained with the endoscopic scraper was significantly higher than when using the brush (1917 vs. 1014 cells, *p* = 0.042). Furthermore, seven cases (8.3%; 7/84) were diagnosed by immunohistochemical analysis of the cell block section obtained from the endoscopic scraper. Given its simplicity and greater capacity for sample acquisition, use of the endoscopic scraper in conjunction with the cell block technique could replace brush cytology for the histological diagnosis of malignant biliary strictures.

## 1. Introduction

Despite advances in various imaging modalities, such as multidetector computed tomography, magnetic resonance cholangiopancreatography, and endoscopic ultrasonography, accurate diagnosis of biliary strictures based on imaging findings continues to be challenging. Therefore, it is essential to obtain samples during endoscopic retrograde cholangiopancreatography (ERCP) to determine the underlying cause of the biliary strictures on the basis of histological evidence. Bile aspiration cytology, brush cytology, and forceps biopsy are conventionally used to obtain specimens for pathological diagnosis [1,2,3,4]. Recently, peroral cholangioscopy (POCS)-guided forceps biopsy has been performed by several groups [5,6]. However, brush cytology is still the most commonly used method worldwide, despite its low sensitivity and accuracy, as it is technically easy to implement, rapid, and cost-effective. On the other hand, forceps biopsy is technically more difficult and time-consuming than brush cytology [3,7]; moreover, POCS-guided forceps biopsy is costly, although it is also precise and highly sensitive.

An endoscopic scraper (Trefle; Piolax Medical Devices, Yokohama, Japan; Appendix A) is now available [8] and has been used for the diagnosis of biliary strictures, mainly in Japan [9,10]. This device has unique configurations; the wire-guided system is designed to access biliary strictures over the guidewire, and three scraping loops are used specifically to obtain tissues and cell samples for histology and cytology. The procedure using the endoscopic scraper is nearly identical to that for the conventional brush; thus, the method is technically easy and rapid. However, this scraper device has yet to gain popularity on a global scale.

This endoscopic scraper collects scraped tissues and cells through the side port of the outer sheath into a syringe, aspirating together with bile juice. With the original method [8], obtained specimens, including tissues and bile juice, are divided into tissue and fluid components for histological and cytological analyses. However, in a clinical site, tissue discrimination is often hampered by the opacity of the surrounding fluid. Therefore, in our facility, the cell block technique is typically used as a simpler method of sample processing, in which it is not necessary to separate the tissue and fluid components [11]. In addition, the cell block method is appropriate for immunohistochemical analysis for differential diagnosis, as well as for distinguishing between benignancy and malignancy [12,13,14]. 

The purpose of this study was to compare the diagnostic yield for biliary strictures between an endoscopic scraper with the cell block method and conventional brush cytology with and without the cell block method, by analyzing a large retrospective sample. Furthermore, we quantified the cells in the cell block section obtained by the scraper and brush using quantitative digital image analysis.

## 2. Patients and Methods

### 2.1. Patients

Between February 2005 and October 2021, 435 consecutive patients with biliary strictures underwent endoscopic transpapillary brush cytology, brush cell block, or forceps biopsy at the Nagoya City University Hospital. Our hospital started using the endoscopic scraper and cell block method in April 2017. The final diagnosis was confirmed based on histological analysis of surgical specimens, other histological findings, imaging data, and the clinical course. The study was approved by the Review Board of Nagoya City University Graduate School of Medical Sciences (approval no. 60-18-0022). Written informed consent was obtained from all patients.

### 2.2. Endoscopic Procedure

ERCP was performed in patients under conscious sedation. After biliary cannulation and the acquisition of a sufficient contrast-filled image, a 0.025-inch guidewire was advanced into the intrahepatic bile duct through the biliary stricture. The conventional brush was inserted beyond the stricture over the guidewire, and moved to and fro across the stricture approximately 10 times. The forceps reached the stricture, with and without the biliary introducer, under X-ray fluoroscopy, and 2–3 biopsied specimens were obtained. The endoscopic scraper (Trefle) was inserted into the bile duct and passed through the stricture over the guidewire, in the same way as for the conventional brush. Then, the opened scraping loops were pulled down in the proximal-to-distal direction through the stricture under fluoroscopic guidance. Tissues and/or cell samples were scraped and then obtained, together with bile juice, by aspiration from the side port of the outer sheath into a syringe. This process was repeated 3–5 times. All specimens, including aspirated bile juice and tissues, were transferred to a sterile tube from the syringe; the scraping loops were cut using scissors into the same sterile tube. Furthermore, bile juice left inside tube was pushed out by air (Appendix A). The tube was submitted to the Pathology Department for the cell block preparation, which enables differentiation of benign from malignant lesions, as well as immunohistochemical analysis (as needed).

### 2.3. Pathological Examination

For cell block preparation, the centrifuged samples were fixed in formalin overnight. Then, they were washed in saline, mixed with 1% sodium aspartate, and centrifuged again. Finally, a few drops of 1 M calcium chloride were added to each sample. The samples were then embedded in paraffin to yield cell blocks. Each cell block was sectioned for hematoxylin-and-eosin (H&E) staining and examined by experienced pathologists (H.K. and S.T.), who classified the samples into five categories: inadequate, benign, atypical, suspicious for malignancy, and malignant. The specimens classified as suspicious for malignancy or malignant were deemed positive. Brush cytological samples and forceps biopsy samples were evaluated in the same manner.

### 2.4. Quantitative Digital Image Analysis

Digital image analysis was performed on cell block sections obtained using a conventional brush and an endoscopic scraper. To count the number of epithelial cells on the diagnosed cell block slides, all slides were taken in by Apreio CS2 (Leica Biosystems, Nussloch, Germany). As shown in Figure 1, only areas with epithelial components were selected. Furthermore, inflammatory cells and stromal cells were excluded based on the size and shape of the nuclei; only the nuclei of the epithelial component were measured. The same parameters used to estimate the number of epithelial cells in the entire cell block specimen (collected at one time) were applied to all individual slides.

### 2.5. Statistical Analysis

Analyses were performed using SPSS software version 21(IBM, Armonk, NY, USA). The χ2 test, Fisher’s exact test, and Mann–Whitney U-test were used for statistical analysis where appropriate. All tests were two-sided, and *p* < 0.05 were regarded as statistically significant. 

## 3. Results

### 3.1. Patient Characteristics

The clinical characteristics of all patients are summarized in Table 1. A total of 435 consecutive patients (301 males and 134 females; median age 73 years) were included in this study. Malignant disease was confirmed in 75.4% (328/435) of the patients; among whom 137 had bile duct cancer, 125 had pancreatic cancer, 28 had gallbladder cancer, and 38 had other types of cancer. Among the 24.6% (107/435) of patients with benign disease, 50 had immunoglobulin G4-related sclerosing cholangitis, 15 had primary sclerosing cholangitis, and 42 had other disease types.

The endoscopic scraper (Trefle) was used in 101 (23.2%) patients. All specimens in the scraper group were processed using the cell block technique. Meanwhile, of the 243 specimens obtained using a conventional brush, 188 (43.2%) underwent a cytology evaluation and 55 (12.6%) were evaluated using the cell block technique. Forceps biopsy was performed in 326 (74.9%) patients.

### 3.2. Diagnostic Yield of the Sampling Methods

Table 2 summarizes the diagnostic yield of the endoscopic scraper with the cell block technique (“Scraper cell block”), conventional brush cytology (“Brush cytology”), conventional brush with the cell block technique (“Brush cell block”), and forceps biopsy (“Forceps biopsy”). We evaluated the sensitivity, specificity, positive predictive value (PPV), negative predictive value (NPV), and accuracy of each sampling method. The sensitivity of the scraper cell block was significantly higher than that of brush cytology and the brush cell block for all malignant diseases (53.6% vs. 30.9%, *p* < 0.001; 53.6% vs. 31.6%, *p* = 0.024, respectively). The sensitivity of forceps biopsy was higher than that of the brush cell block (62.3% vs. 53.6%; *p* = 0.157), but not significantly.

### 3.3. Comparison of Cell Counts in the Cell Block Section between Endoscopic Scraper and Conventional Brush

To compare the number of cells obtained using the endoscopic scraper and conventional brush, quantitative digital image analysis of each cell block section was performed in the scraper cell block and brush cell block groups. The clinical characteristics of the patients diagnosed with a malignancy in the scraper cell block and brush cell block groups are shown in Table 3. There was no obvious difference between the two groups in terms of the final diagnosis. Nuclei were counted only in the epithelial component; inflammatory and stromal cells were excluded (Figure 1A–D). Figure 1E shows that the median number of cells obtained by the scraper was significantly higher than that obtained with the brush (1917 vs. 1014 cells; *p* = 0.042).

### 3.4. Utility of Immunohistochemistry for Differential Diagnosis

Immunohistochemical analyses can be conducted on cell block sections to narrow the differential diagnosis. In this study, 7 of 84 cases (8.3%) were diagnosed based on the immunohistochemistry of the cell block section from the scraper cell block group, as shown in Table 4. Representative immunohistochemistry images are shown in Figure 2. In the case shown in Figure 2A, determination of cytokeratin 7 (CK 7), CK 20, and caudal-type homeobox 2 (CDX 2) expression was useful for diagnosing colon cancer. The negative findings for CK 7 and positive findings for CK 20 and CDX 2 indicate a final diagnosis of pancreatic metastasis from colon cancer. The case in Figure 2B was diagnosed with neuroendocrine carcinoma based on positive immunostaining of CD56, chromogranin, and synaptophysin. The case in Figure 2C was positive for GATA3 and p40, and focally positive for CK20; this supported a diagnosis of lymph node metastasis from urothelial cancer.

## 4. Discussion

Histological data must be obtained prior to use of invasive therapies, such as surgery or chemotherapy, for malignant biliary strictures. In a systematic review and meta-analysis, the pooled sensitivities of brush cytology and forceps biopsy were 45% and 48.1%, respectively [3]. Some reports have indicated that forceps biopsy has higher sensitivity for malignancy (50–60%) than brush cytology, with a specificity of 96–98% [15,16]. Although endoscopic transpapillary forceps biopsy can be used to obtain larger tissue samples, its success is dependent on operator skill, because it is technically more difficult to insert thick forceps into the bile duct to grasp the target lesion. Therefore, brush cytology remains the primary method in many facilities for obtaining specimens from biliary strictures, despite its low sensitivity, because it is technically easy, rapid, and cost-effective. To improve sensitivity for diagnosing pancreatic cancer, it is known that endoscopic-ultrasound-guided fine-needle aspiration/biopsy (EUS-FNA/FNB) is a standard method. On the other hand, in order to improve sensitivity for existing malignant biliary strictures, except for pancreatic cancer, various sampling strategies have been evaluated, mainly at high-volume centers, such as EUS-FNA/FNB [17,18,19], POCS-guided forceps biopsy [20,21,22], probe-based confocal laser endomicroscopy (pCLE) [23,24,25], fluorescence in situ hybridization [2,26,27], and intraductal aspiration as a “scraping-based” technique using the tip of a conventional brush catheter [28]. These methods have higher sensitivity and accuracy for diagnosing biliary strictures than brush cytology and forceps biopsy. However, they have not been widely applied due to the requirements for specific technical skills or costly equipment. Hence, alternatives with technically easy, rapid, and cost-effective method are required.

The endoscopic scraper (Trefle) has several unique features. One such feature is its wire-guided system, which can access biliary strictures, including intrahepatic lesions, for easy and rapid sample acquisition [8]. Another unique feature is its three metallic loops, which yield tissue samples of an adequate size [8]. Therefore, we speculate that this endoscopic scraper could replace brush cytology for diagnosing malignant biliary strictures. However, to our knowledge, the size of tissue samples obtained using the endoscopic scraper has not been compared to that obtained by brush cytology. Thus, we compared the number of cells on the cell block section between the scraper and conventional brush, as cytological specimens were unavailable for obtaining cell counts. Quantitative digital image analysis revealed that the number of cells obtained from the scraper cell block group was significantly higher than that from the brush cell block group. 

Although the sensitivity of the scraper cell block was significantly higher than that of brush cytology and the brush cell block in the current study, the diagnostic yields of the scraper cell block were still insufficient and low compared to the previous reports. In general, it is difficult to compare the sensitivity and specificity among different facilities. However, we emphasize the superiority of the scraper over the conventional brush, in terms of the sensitivity and the number of cells collected, for a single facility and the same sample processing.

A prospective randomized controlled study reported higher sensitivity for bile cytology with the cell block method than conventional smear cytology [13]. Use of sodium aspartate as a fixative increases cellularity, which in turn increases morphological detail and diagnostic sensitivity. However, our data are not consistent with this report; there was no difference in sensitivity between brush cytology and the brush cell block. The advantage of a combination with the cell block method is that multiple sections can be generated (as needed) for staining and immunohistochemistry [29], whereas cytology specimens are unsuitable for immunohistochemical analysis. In addition, having a larger number of cells should facilitate immunohistochemical analysis; as such, the endoscopic scraper is a more promising device for sample acquisition than the conventional brush. Furthermore, in terms of adverse events, there was no case in which massive bleeding occurred immediately after the sampling procedure of the endoscopic scraper.

In the future, diagnosis and treatment of biliary tract cancers should be based on molecular and genetic analyses of specimens obtained during ERCP from patients with unresectable biliary tract cancers, given the recent developments in chemotherapy. The s of DNA necessary for next-generation sequencing depends on the platform, gene panel size, and enrichment process [30]. To test for “hotspots” in 50 genes using the multiplex PCR-based Ion AmpliSeq Cancer Hotspot Panel (Life Technologies, Carlsbad, CA, USA), 10 ng of input DNA is required, which corresponds to approximately 2000 target cells [31]. In the present study, the mean number of cells obtained by the endoscopic scraper was approximately 2000. On the basis of this result, specimens obtained using the endoscopic scraper could be subjected to gene panel evaluation, only in cases in which an adequate number of cells were collected. However, further investigations are required, as recent gene panels with hundreds of genes require a greater DNA volume [30].

Although the present study covered a long period, in our hospital, the endoscopic procedure and handling the specimens did not change during the survey period. However, this study had several limitations, including its retrospective, single-center design, moderate number of patients, and potential bias arising from discrepant case numbers. Therefore, a prospective, randomized controlled trial involving a larger number of patients is required to confirm our results.

## 5. Conclusions

In conclusion, our study revealed that use of the endoscopic scraper and cell block method achieved higher sensitivity than brush cytology and use of a brush with the cell block method for the diagnosis of malignant biliary strictures. Furthermore, quantitative digital image analysis showed that the number of cells on the cell block sections obtained by the endoscopic scraper were significantly higher than those obtained using the conventional brush. Given its ease of use and high sample acquisition capability, the endoscopic scraper might replace brush cytology for diagnosing malignant biliary strictures.

## Figures and Tables

**Figure 1 cancers-14-04147-f001:**
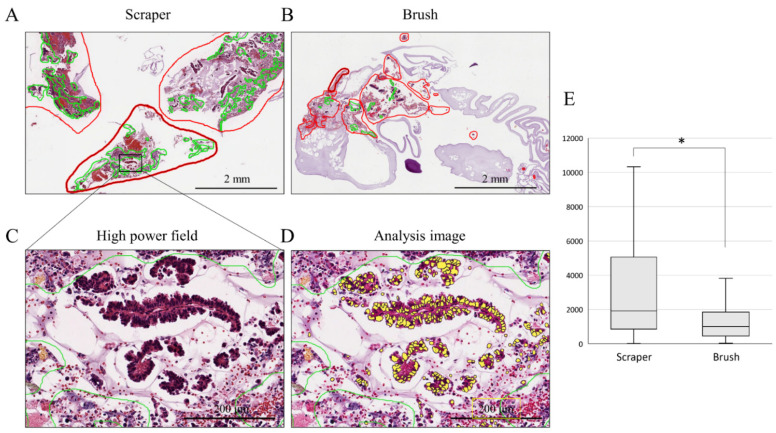
Epithelial cell count analysis on cell block sections. (**A**,**B**): histological findings of scraper group (**A**) and brush group (**B**) in low-power field. The areas with epithelial components are circled in red. The areas circled by the green line were excluded due to the presence of necrotic tissue and inflammatory cells. (**C**,**D**): histological findings in high-power field (**C**) and analysis image of Aperio CS2 (**D**). Yellow-colored nuclei; only the nuclei of the epithelial component were measured. (**E**): comparison of the median numbers of cells obtained by each device; *, *p* < 0.05.

**Figure 2 cancers-14-04147-f002:**
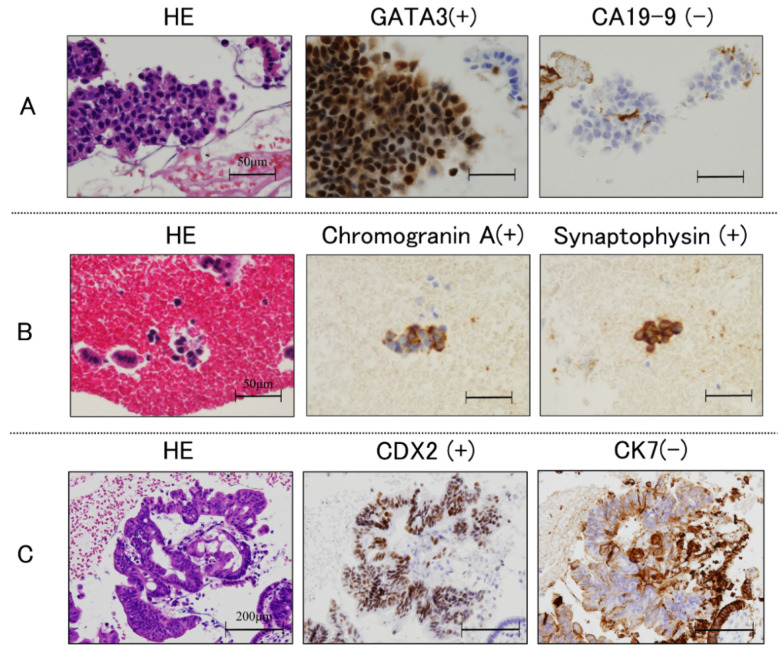
Histological findings. Immunohistochemical staining of cell block sections obtained using endoscopic scraper; (**A**): negative for cytokeratin 7 (CK 7), and positive for CK 20 and caudal-type homeobox 2 (CDX 2); (**B**): positive for CD56, chromogranin and synaptophysin; (**C**): positive for GATA3 and p40, and focally positive for CK20.

**Table 1 cancers-14-04147-t001:** Clinical characteristics.

Characteristic of the Patients (*N* = 435)	
Sex, *n* (Male/Female)	301/134
Age (years), median (range)	73 (23–98)
Final diagnosis	
Malignant disease, *n* (%)	328 (75.4%)
Bile duct cancer, *n*	137
Pancreatic cancer, *n*	125
Gallbladder cancer, *n*	28
Others, *n*	38
Benign disease, *n* (%)	107 (24.6%)
IgG4-related sclerosing cholangitis, *n*	50
Primary sclerosing cholangitis, *n*	15
Others, *n*	42
Sampling method	
Scraper (Trefle) + cell block, *n* (%)	101 (23.2%)
Brush cytology, *n* (%)	188 (43.2%)
Brush + cell block, *n* (%)	55 (12.6%)
Forceps biopsy, *n* (%)	326 (74.9%)

**Table 2 cancers-14-04147-t002:** Diagnostic yield of each sampling method.

	Scraper Cell Block	Brush Cytology	Brush Cell Block	Forceps Biopsy
Sensitivity % (*n*/*N*)	53.6 (45/84)	30.9 (46/149) ^a^ *p* = 0.0006	31.6 (12/38) ^a^ *p* = 0.024	62.3 (157/252) ^a^ *p* = 0.157
Bile duct cancer	65.5 (19/29)	34.4 (21/61) ^a^ *p* = 0.0055	38.9 (7/18) ^a^ *p* = 0.074	76.3 (87/114) ^a^ *p* = 0.236
Pancreatic cancer	41.2 (14/34)	29.0 (18/62) ^a^ *p* = 0.227	27.3 (3/11) ^a^ *p* = 0.408	46.6 (41/88) ^a^ *p* = 0.59
Specificity % (*n*/*N*)	100 (17/17)	97.4 (38/39)	100 (17/17)	100 (74/74)
PPV % (*n*/*N*)	100 (45/45)	97.9 (46/47)	100 (12/12)	100 (157/157)
NPV % (*n*/*N*)	30.4 (17/56)	27.0 (38/141)	39.5 (17/43)	43.8 (74/169)
Accuracy % (*n*/*N*)	61.4 (62/101)	44.7 (84/188) ^a^ *p* = 0.0068	52.7 (29/55) ^a^ *p* = 0.295	70.9 (231/326) ^a^ *p* = 0.073

PPV—positive predictive value, NPV—negative predictive value. ^a^—*p* value compared with scraper cell block group.

**Table 3 cancers-14-04147-t003:** Clinical characteristics of the patients diagnosed malignancy by scraper cell block and brush cell block.

Characteristic of the Patients	Scraper Cell Block (*n* = 45)	Brush Cell Block (*n* = 12)
Sex, *n* (Male/Female)	28/17	8/4
Age (years), median (range)	76 (53–98)	64 (51–89)
Final diagnosis		
Bile duct cancer, *n* (%)	19 (42)	7 (58)
Hilar/Distal, *n*	7/12	3/4
Pancreatic cancer, *n* (%)	14 (31)	3 (25)
Gallbladder cancer, *n* (%)	6 (13)	2 (17)
Others, *n* (%)	6 (13)	0 (0)

**Table 4 cancers-14-04147-t004:** Cases in which immunohistochemistry was helpful for differential diagnosis.

#	Age	Sex	Final Diagnosis	Immunohistochemical Staining	Excludable	Figure 2
1	73	M	Bile duct cancer	CK7(+), CK20(+), CK19(+), CEA(+) CK5/6(+/−), p40(−)	Metastasis from pharyngeal cancer	
2	81	M	Pancreatic cancer	CK7(+), CK20(+), CA19-9(+), MUC1(+), TTF1(−)	Metastasis from lung cancer	
3	98	F	Gallbladder cancer	CK7(+), CK20(−), CA19-9(+) GCDFP15(−), Mammaglobin(−), ER(+/−),	Metastasis from breast cancer	
4	75	M	Lymph node metastasis from urothelial carcinoma	GATA3(+), p40(+) CA19-9(−), MUC1(+/−)	Bile duct cancer	A
5	66	M	Neuroendocrine tumor	CD56(+), Chromogranin A(+), Synaptophysin(+), CA19-9(−)	Bile duct cancer	B
6	53	M	Pancreatic metastasis from colon cancer	CK7(−), CK20(+), CDX2(+)	Pancreatic cancer	C
7	72	M	Sarcomatoid hepatocellular carcinoma	CKAE1/3(+), Vimentin(+), Hepatocyte(−), CK19(−)	Bile duct cancer	

## Data Availability

The data presented in this study are available from the corresponding authors on reasonable request.

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
