# Peer review of "Use of Endoscopic Scraper and Cell Block Technique as a Replacement for Conventional Brush for Diagnosing Malignant Biliary Strictures"

_cancers, 2022, doi:10.3390/cancers14174147_

Round 1

Reviewer 1 Report

Kato et al describe a retrospective study of an interesting instrument to improve a clinical challenge for classifying biliary strictures.

Some questions/remarks remain:

Please add an image of endoscopic scraper.

This is a very long period to look at. Did the procedure and handling of the specimens not change over time?

Yield of brush/ intraductal collection of samples for pancreatic carcinoma is known to be lower than for biliary cancers. For pancreatic cancers EUS FNA is the gold standard. Please discuss.

Yields of brush and scraper are low when compared to current literature. Please discuss.

Any difference in adverse events? Pancreaitits? Cholangitis? Bleeding?

The conclusion should be weakend. More prospective data is needed.

Author Response

Responses to the comments of Reviewer 1

  1. Please add an image of endoscopic scraper.

We agree that this image is important. Therefore, an image of endoscopic scraper was shown as Supplemental Figure 1. However, we forgot to cite Figure S1 in the maintext, resulting in confusion of the reviewer. We added a citation for Figure S1 showing an image of scraper as indicated below;

An endoscopic scraper (Trefle; Piolax Medical Devices, Yokohama, Japan; Supplementary Figure 1) is now available[8]

  1. This is a very long period to look at. Did the procedure and handling of the specimens not change over time?

As the reviewer indicated, this study was the retrospective study summarized between 2005 and 2021. That is a very long period to investigate. However, I.N., corresponding author for this article, has been a leader of endoscopist in our hospital during those periods. Likewise, S.T. has been a professor of department of pathology. Therefore, the endoscopic procedure and handling of the specimens did not change during those periods.

  1. Yield of brush/ intraductal collection of samples for pancreatic carcinoma is known to be lower than for biliary cancers. For pancreatic cancers EUS FNA is the gold standard. Please discuss.

Thank you for your valuable suggestions on our manuscript. We agree that EUS-FNA is the gold standard to obtain histological evidence for pancreatic cancer. According to the reviewer’s comment, we added the description about EUS-FNA against biliary strictures caused pancreatic cancer in Discussion part as shown below;

To improve sensitivity for diagnosing pancreatic cancer, it is known that endoscopic ultrasound-guided fine needle aspiration/biopsy (EUS-FNA/FNB) is a standard method. On the other hand, in order to improve sensitivity for malignant biliary strictures caused except for pancreatic cancer, various sampling strategies have been evaluated, mainly at the high volume centers, such as…

  1. Yields of brush and scraper are low when compared to current literature. Please discuss.

We welcome the reviewer’s indication, and agree that the diagnostic yields of Brush and Scraper in our present study were still low compared to the previous reports. However, in general, it is difficult to compare the sensitivity and specificity among different facilities, because the pathologists are different. Our most urgent message is the superiority of the scraper over the brush, in a single facility and the same sample processing. Therefore, the description for this point was modified in Discussion section as indicated below;

the diagnostic yields of Scraper cell block were still insufficient and low compared to the previous reports. In general, it is difficult to compare the sensitivity and specificity among different facilities. However, we emphasize the superiority of the scraper over the conventional brush, in terms of the sensitivity and the amounts of cells collected, for a single facility and the same sample processing.

  1. Any difference in adverse events? Pancreaitits? Cholangitis? Bleeding?

We thank the reviewer for this pertinent comment and agree that this information is useful for readers. However, it is difficult to compare the rate of adverse events among each method, because there were many cases in which combination of sampling method were performed. At least, there was no case in which massive bleeding occurred immediately after sampling procedure of endoscopic scraper.

  1. The conclusion should be weakend. More prospective data is needed.

We appreciate the reviewer’s indication. As suggested, we modified the conclusion as described below;

Given its ease of use and high sample acquisition capability, the endoscopic scraper might replace brush cytology for diagnosing malignant biliary strictures.

Reviewer 2 Report

Intraductal aspiration has been previously described by Curcio et.al as a “scraping-based” technique using the tip of a standard brushing catheter to improve the diagnostic yield in the assessment of biliary strictures.

(Curcio G, Traina M, Mocciaro F, Liotta R, Gentile R, Tarantino I, Barresi L, Granata A, Tuzzolino F, Gridelli B. Intraductal aspiration: a promising new tissue-sampling technique for the diagnosis of suspected malignant biliary strictures. Gastrointest Endosc. 2012 Apr;75(4):798-804. doi: 10.1016/j.gie.2011.12.005. Epub 2012 Jan 31. PMID: 22301344.)

Please add the previous comment with reference in the Discussion section.

Author Response

Responses to the comments of Reviewer 2

Please add the previous comment with reference in the Discussion section.

Thank you for your valuable suggestions on our manuscript. As suggested, we added the comments with reference in Discussion section as described below;

On the other hand, in order to improve sensitivity for malignant biliary strictures caused except for pancreatic cancer, various sampling strategies have been evaluated, mainly at the high volume centers, such as EUS-FNA/FNB[17-19], POCS-guided forceps biopsy[20-22], probe-based confocal laser endomicroscopy (pCLE)[23-25], fluorescence in situ hybridization[2,26,27], and intraductal aspiration as a “scraping-based” technique using the tip of a conventional brush catheter[28].

Round 2

Reviewer 1 Report

please add discussion of comment number 2 and 5 to the discussion

Author Response

Responses to the comment of Reviewer 1 (R2)

・Please add discussion of comment number 2 and 5 to the discussion

  1. This is a very long period to look at. Did the procedure and handling of the specimens not change over time?

As the reviewer indicated, this study was the retrospective study summarized between 2005 and 2021. That is a very long period to investigate. However, I.N., corresponding author for this article, has been a leader of endoscopist in our hospital during those periods. Likewise, S.T. has been a professor of department of pathology. Therefore, the endoscopic procedure and handling of the specimens did not change during those periods. We added the comments about this point in Discussion section as described below;

The present study summarized for a long period, whereas, in our hospital, the endoscopic procedure and handling the specimens did not change during survey period. However, this study had several limitations, including its retrospective, single-center design, moderate number of patients,

  1. Any difference in adverse events? Pancreaitits? Cholangitis? Bleeding?

We thank the reviewer for this pertinent comment and agree that this information is useful for readers. However, it is difficult to compare the rate of adverse events among each method, because there were many cases in which combination of sampling method were performed. At least, there was no case in which massive bleeding occurred immediately after sampling procedure of endoscopic scraper. Therefore, we added the comments about this point in Discussion section as described below;

In addition, having a larger amount of cells should facilitate immunohistochemical analysis; as such, the endoscopic scraper is a promising device for sample acquisition than conventional brush. Furthermore, in terms of adverse events, there was no case in which massive bleeding occurred immediately after sampling procedure of endoscopic scraper.